# Particulate Air Pollution and Primary Care Visits in Kosovo: A Time-Series Approach

**DOI:** 10.3390/ijerph192416591

**Published:** 2022-12-09

**Authors:** Zana Shabani Isenaj, Merita Berisha, Antigona Ukëhaxhaj, Hanns Moshammer

**Affiliations:** 1Medical Faculty, University of Hasan Prishtina, Rr. George Bush Nr. 31, 10000 Pristina, Kosovo; 2National Institute of Public Health, St. Mother Teresa pn, Rrethi i Spitalit, 10000 Pristina, Kosovo; 3Master Programme, Medical Faculty, University Fehmi Agani, Rr. Ismail Qemali n.n., 50000 Gjakova, Kosovo; 4Department of Environmental Health, ZPH, Medical University of Vienna, 1090 Vienna, Austria; 5Department of Hygiene, Medical University of Karakalpakstan, Nukus 230100, Uzbekistan

**Keywords:** respiratory health, cardiovascular health, primary care visits, particulate air pollution

## Abstract

This study aimed to investigate the effects of particulate air pollution (PM2.5) on cardiovascular and respiratory diseases in Pristina, Kosovo, in a time-series analysis using daily primary healthcare visits to primary care institutions from 2019 to 2022. For the observation period, 6440 cardiovascular and 15,141 respiratory visits were reported, whereas the daily mean concentrations of PM2.5 ranged between 2.41 and 120.3 µg/m^3^. Single-lag models indicated a bi-phasic lag structure with increasing effect estimates some days after the air pollution event. In the distributed lag model with seven lags, the effect estimates for the cardiovascular cases indicated the adverse effect of air pollution. The cumulative effect estimate (summed over lag 0 to 6) for an increase of 10 µg/m^3^ of PM2.5 was a relative risk of 1.010 (95% confidence interval: 1.001–1.019). For respiratory cases, a different lag model (lag 4 through 10) was additionally examined. In this model, significant increases in visits were observed on lags 7 and 8. Overall, no relevant increase in visits occurred during the seven days considered. Visits to general practitioners will often not occur immediately at disease onset because patients will wait, hoping that their health status improves spontaneously. Therefore, we expected some latency in the effects.

## 1. Introduction

Air pollution is a well-established risk factor with a severe negative health impact [1,2]. Airborne pollutants directly reach the airways and therefore they cause direct damage to the respiratory tract [3]. Especially for particulate air pollution, oxidative stress and inflammatory responses are well-known mechanisms of effect. These mechanisms also have systemic consequences among which cardiovascular pathologies are the most important because of their high prevalence [4] and have been described for many years in consensus reports [5,6]. Certainly, there are spatial differences in air pollution depending on the distance to the local pollution sources. However, in an area as large as a big city, basin or valley between mountains, the concentrations of pollutants at different monitoring stations are usually highly correlated with each other over time. Therefore, daily concentrations at a well-selected monitor provide sufficient information about daily variation of exposure in a large number of people. The measurement error is relatively small, likely non-differential, and offset by an often large study population.

In order to reap the full advantage of the large study population, health data are usually taken from official registers. Cause of death and hospital admission data are most often used in time-series studies on the short-term health effects of air pollution. Regarding mortality, effect estimates are usually below a 1 percent increase per 10 µg/m^3^ PM10 [7,8,9] and around 1 percent for PM2.5 [10]. Such effect sizes are small and can only be detected in a sufficiently large population. It can be rightfully criticized that these mortality risks are not representative of the health risk of the population at large, but only describe the increased risk of the highly vulnerable subpopulation of very old or very sick patients that have an increased risk of death anyways. Hospital admission data are used to broaden the representativeness of the time-series study [11] with similar effect estimates as for mortality. We have recently performed such a study on hospital admissions, analyzing admissions to the pediatric clinic in Pristina, Kosovo [12]. Air pollution effects on the hospital admission data also strongly depend on the type of admission [13], which is not always available in the register data. As we have discussed in our paper [12], the source population of a single hospital is not well defined and therefore, a clear interpretation of the results regarding population health is difficult. Routine data from primary care reported by resident doctors would overcome some of these difficulties, as these data would cover the whole population. As opposed to the mortality and hospital admission data, the primary care data are only rarely available in sufficient quality and have therefore not been used as extensively in time-series studies on air pollution health effects. A new electronic reporting system for all primary healthcare providers in Kosovo was established in 2019 and we decided to make use of this dataset beginning on 23 September 2019, when we deemed that the reporting was sufficiently well established.

## 2. Materials and Methods

Here, we report the results of a time-series study on the effects of daily variation in particulate matter (PM2.5) concentration on the number of primary care visits in the municipality of Pristina, Kosovo. In our previous study [12], we demonstrated that PM2.5 data from the central monitoring station in Pristina display a strong temporal correlation with all of the other monitoring stations in the inhabited areas of Kosovo. Therefore, these data were deemed representative for the whole country. The PM2.5 monitoring was performed using a Grimm Environmental Dust Monitor EDM 180 [14]. As described in the previous study, rare data gaps were closed by using the data from the US Embassy, applying the formula derived from linear regression.

All general practitioners and other resident doctors are obliged to report each patient visit daily, reporting date of visit, gender, birth date and main diagnosis using the WHO International Statistical Classification of Diseases and Related Health Problems, 10th Revision (ICD10). Type of medical institution (specialty) and municipality are also reported. For this pilot study, we restricted the analysis to medical institutions from Pristina. We restricted our study to the observation period from 23 September 2019 to 20 February 2022, and to cardiovascular and respiratory diagnoses. For cardiovascular diagnoses, all ICD10-codes starting with “I” were included. For respiratory diagnoses, we included all ICD-codes starting with “J”. Realizing that COVID-19 would also mostly present itself as a respiratory disease and diagnostic coding would depend on testing capacity and thus change over time, we also included the novel code U.07 for COVID-19 among the respiratory diagnoses.

We built general additive models (GAM), family: Poisson, in the same way as described in our previous study [12] following standard procedures [15,16,17,18,19,20,21]: the number of knots of the spline for the temporal variation was selected based on the minimal residual autocorrelation. The best lag for the temperature and for the temperature on consecutive days was chosen based on the Akaike Information Criterion (AIC) [22]. Also, the day of the week was included as a factor variable. Calculations were conducted using R [23].

Most visits were with general practitioners. Unlike emergency hospital admissions, most visits to general practitioners do not concern immediately life-threatening health states. Therefore, often patients that observe symptoms will not go to the doctor immediately but will first wait and see if their health status improves spontaneously. Therefore, we expected some latency between the air pollution episode and the visit. At first, we examined air pollution effects after different lags separately. Pending confirmation by visual inspection of our a priori hypothesis that the lag structure resembled a third-degree polynomial indicating some “harvesting” effect [12], a third-degree polynomial distributed lag model accounting for seven days (lag 0–lag 6) was implemented. 

## 3. Results

In the observation period, 6440 cardiovascular (Table 1) and 15,141 respiratory visits (11,344 with a “J” and 3797 with a “U07” diagnosis, Table 2) were reported. The particulate matter (PM2.5) concentration ranged from 2.4 to 120.3 µg/m^3^ (arithmetic mean: 21.8), temperature from −8.6 to 28.3 °C (mean: 10.8 °C). For cardiovascular and respiratory cases, a spline for the temporal trend with 25 and 23 knots, respectively, fared best.

### Time-Series Results

For both outcomes, the single lag models indicated a bi-phasic lag structure with increasing effect estimates four or five days after the air pollution event. In the distributed lag model with 7 lags (Figure 1), the effect estimates of the respiratory cases were clearly negative (indicating fewer visits with increasing pollution levels) on days 0 to 3. On the following days, the adverse effect estimates did not reach significance. Conversely, slightly negative effects in the first days (lag 1 and 2) were not significant in the cardiovascular cases. Later on, however, the effect estimates were significantly positive, indicating the adverse effect of air pollution. The cumulative effect estimate for the cardiovascular visits (summed over lag 0 to 6) for an increase of 10 µg/m^3^ of PM2.5 was a relative risk of 1.010 (95% confidence interval: 1.001–1.019).

In an exploratory analysis, we also examined PM2.5 effects on the respiratory visits on lags 4 through 10. To that end, we simply moved the visit counts up by four days in the data table and then ran the same model. Likely because of the somewhat shorter observation period, a spline for the temporal trend with 22 knots resulted in the smallest residual autocorrelation. Figure 2 shows that in this model, respiratory visits increased significantly on lags 7 and 8. Overall, no clear increase or decrease in visits was observed during the seven-day period. Per 10 µg/m^3^ increase, there was an overall increase in visits of 0.02% (95% confidence interval: −0.52%, 0.56%).

## 4. Discussion

Although many studies have examined the short-term effects of air pollution on health outcomes, the data from primary care have only rarely been examined [24]. An electronic database of primary care visits offers a wide range of opportunities for research and also supports informed policy decisions. This pilot study took advantage of these newly available high-quality data and at the same time examined the availability and usefulness of the new electronic database from Kosovo. The analysis was restricted to a relatively short time interval of about 2.5 years and only in the municipality of Pristina. This is a rather short observation period for a time-series analysis. Furthermore, we cannot be sure that all practitioners consistently adhered to the reporting requirements. However, missing reports are likely non-differential regarding the daily air pollution levels. Therefore, missing data might undermine the representativeness of the data (especially if reporting adherence varies across districts or regions) but would not introduce any bias.

The coverage of air pollution data for the whole country is sufficient, but it was not completely clear if adherence to the new reporting rules was equally well developed in more remote parts of the country, especially in the introductory phase of the system. With a more complete establishment of the reporting system, country-wide analyses including time-series approaches to look at short-term temporal variation, as well as spatial analyses, will become possible though.

The effect estimates found for cardiovascular visits were of plausible magnitude [10,25]. Also, a lag time of a few days between an air pollution episode and visit to the doctor was not unexpected. In primary health care, most visits are not because of a life-threatening event. In the case of a less severe disease, patients will usually wait a few days and would only consult their doctor if their health status did not improve.

The seemingly immediate protective effect of air pollution reducing the respiratory visits was unexpected. Nevertheless, after a latency of about five days, the adverse effects nearly reached significance. In rural Austria, regarding respiratory hospital admissions, we even observed a lag time of even 10 (males) or 11 (females) days [26]. In primary health care, most respiratory diseases are likely due to respiratory infections. This would have been even more pronounced during the COVID-19 pandemic when the data of this study were collected.

We have shown previously (with data from Vienna) that air pollution affects infection risk [27]. However, the symptoms that precipitate a consultation with the doctor only occur after an incubation period. For COVID-19, this period is approximately five days [28], which fits our observations very well. Therefore, a different lag model (lag 4 through 10) was additionally examined. In this model, significant increases in the visits were observed on lags 7 and 8. Overall, in all seven days considered, no relevant increase in visits occurred. It must be pointed out that this additional analysis was not planned a priori and therefore was not hypothesis driven. Therefore, a causal interpretation was not appropriate. This finding could rather serve as a basis for new hypotheses (delayed health effects regarding infectious diseases) in future time-series analyses.

It is not completely clear if the new reporting system is fully functional due to adherence to the new reporting rules, thus indicating that there is a higher number of primary care data that are not electronically updated. With a more complete establishment of the reporting system, these limitations will likely be overcome soon.

## 5. Conclusions

The analysis was restricted to a relatively short time interval of only about 2.5 years and to the municipality of Pristina, using data from primary care, which has only rarely been examined. The study found plausible short-term effects of particulate air pollution on cardiovascular morbidity. A delayed effect on respiratory visits requires independent confirmation.

When initial problems with reporting adherence have been solved in the new electronic primary health care reporting system in Kosovo, this system will be an important tool for country-wide analyses including time-series approaches to look at short-term temporal variation, as well as spatial analyses.

## Figures and Tables

**Figure 1 ijerph-19-16591-f001:**
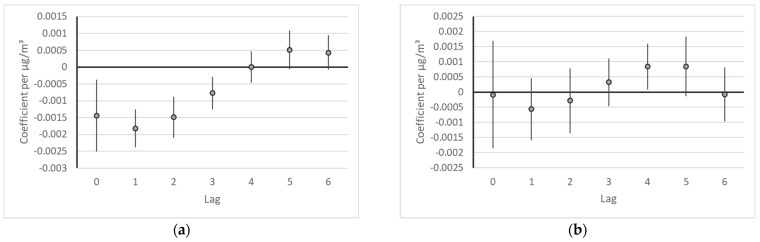
Effect estimates for each lag day from the distributed lag model. Coefficients per µg/m^3^. (**a**) Respiratory visits; (**b**) cardiovascular visits.

**Figure 2 ijerph-19-16591-f002:**
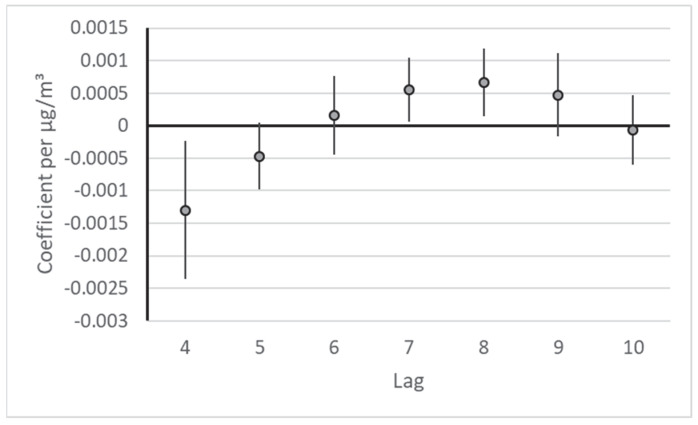
Effect estimates for each lag day from the distributed lag model (lag 4 through 10) for respiratory visits. Coefficients per µg/m^3^.

**Table 1 ijerph-19-16591-t001:** Description of cardiovascular cases.

Gender	Number
Females	3539
Males	2901
**Specialty**	
Family medicine	3949
Internal medicine	2252
Others ^1^	239
Total	6440
Age (mean +/− std. dev.)	61.9 +/− 13.6

^1^ Including rheumatology (113), gynecology (38) and occupational medicine (61).

**Table 2 ijerph-19-16591-t002:** Description of respiratory cases.

Gender	ICD10 = “J”	ICD10 = “U07”
Females	5910	3539
Males	5434	2901
**Specialty**		
Family medicine	8052	3741
Pediatric	1700	3
Occupational medicine	250	14
Internal medicine	197	34
Others	145	5
Total	11,344	3797
Age (mean +/− std. dev.)	29.6 +/− 23.8	41.5 +/− 16.9

## Data Availability

Raw data are available upon request from the corresponding author.

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
