# Peer review of "Particulate Air Pollution and Primary Care Visits in Kosovo: A Time-Series Approach"

_ijerph, 2022, doi:10.3390/ijerph192416591_

Round 1

Reviewer 1 Report

The impact of air quality on health is an increasingly important topic; by analyzing lag times as well as immediate effects this article does capture short-term impacts of air quality that are important for both human health and health-care planning. While this paper is interesting it would be greatly improved by focusing on the important information and reorganizing sections to ensure that the reader can easily understand the flow of the content and that information is correctly placed in the introduction, methods, results, discussion sections. It is also important to characterize the population and exposure more clearly for the readers.

The first paragraph of the introduction focuses on the reason to study air quality over other types of pollution. This is not necessary, and introduces the potential for additional arguments - why can statistical methods not be used to evaluate water and dietary exposures, why is air exposure assumed to be the same for everyone when for many people exposure to outdoor ambient air is limited and they have the option of different protective measures from closing windows to HEPA filtration, etc. It makes more sense to make a case for the importance of outdoor ambient air quality then let it rest. As well, measurement at ground level can be highly variable and quite dependent on proximity to the nearest air quality monitor. The final claim that it is offset by a large study population again is more relevant.

The second paragraph addresses mortality vs morbidity. Again this could be addressed in 1-2 sentences maximum. Additionally, rather than stating that your study only just learned of an EMR (this makes the study sound like good luck rather than good planning), stating that 'the recently implemented EMR provides a new opportunity to evaluate health care data from across Kosovo" which then ties into the "large population" piece from paragraph 1. 

Consider rewriting the introduction as:

Paragraph 1: a broader review of what is known about the impact of air quality events on health, where lag time analyses can also be justified.

Paragraph 2: a broader review of how studies use population health data to analyze air quality impacts on health, what already exists and why this study provides an important contribution to the literature (like completeness due to the new EMR and _____ monitoring stations with ability to do the lag-time analyses). 

Please leave missing data points, length of time, etc to the "limitations" section of the discussion. 

Methods should be divided into:

1. Type of study (observational cohort?) 

2. Air quality data monitoring with a description of how PM2.5 was analyzed. 

3. How the populations were defined and analyzed. You used ICD10 codes specifically pertaining to cardiovascular (____) and respiratory (____) extracted from the EMR, which represents all medical visits conducted in Kosovo over x time period. In the conclusions, 'children' are mentioned. The population for which the data was used needs to be fully characterized in the methods, including the age range.

4. Using previously published models, you analyzed the data. You added additional modelling factors including ____.

The last paragraph in the methods section actually describes results. Other results, like the number of patient visits captured, are also sprinkled throughout the methods section and should be moved. 

Results:

The first paragraph of the results section is helpful but should include some of the population data currently shown in the methods section. Please consider whether a table describing the population would be a helpful way to visualize the overall population description. It feels like there are multiple important areas of discussion including: type of health care visit (GP vs ED/hospital) and cardiovascular vs respiratory, population characteristics (is there access to age and sex data? who goes where, are children or elderly more likely to end up at the hospital?). A little more description of the PM and environmental data would also be helpful; the range and mean of PM data and temperature is provided, but would be more helpful to know as well the # of 'pollution' events, etc; this is not clear from the next section where events are evaluated over time and it is important to know whether there were 1 or 10 being analyzed. 

Discussion

Instead of saying 'this study was mainly meant as a pilot study' consider saying 'This pilot study focused on ... to take advantage of newly available high quality data'.

Consider a bit more comparison between cardiovascular events shown in this study and those in other studies, especially other ones that include lag-time. Also it would be beneficial to have a reference for delayed presentation to primary care as opposed to conjecture.

The conclusions around respiratory events seem overstated. Is this truly protective? Are patients with respiratory complaints attending a more acute care setting or minimizing outdoor air exposure? It seems that there are more options to discuss and this paragraph could be tackled with more detail and exploration. 

The final paragraph of the discussion explores some infection and air pollution risk linkage but it is presented in a confusing way. With ICD-10 codes, it should be possible to differentiate 'all respiratory visits' from 'respiratory visits attributed to infection'; this discussion could be explored with the existing data. As it stands, while there is literature showing that COVID and other respiratory infection outcomes are worse in areas with more air pollution, attribution of respiratory visits during episodes of air pollution to respiratory infection is still an area that would need more exploration and this dataset might be well set up to answer that question.

The limitations of this pilot study can be addressed, as well as describing the deliverable - an analysis system and preliminary data informing best modelling strategies for the incoming much-larger dataset - that this pilot model provides. An additional paragraph looking at limitations and how this analysis sets up future work, just a couple of sentences even, would be a helpful addition.

Conclusions: 

The first time the word "children" is mentioned in in the Conclusions section. It would be important to specify the age group studied in the title, abstract, and introduction and absolutely necessary to describe the age range in the population description in the methods section. Conversely, if this is not children only, that would also be important and useful information. 

The second paragraph considers one of the limitations of the study, a common issue for population-level studies that rely on health records data. This should be addressed in the limitations section that should be included in the discussion. In the conclusions, simply that this limited analysis can be used as a model for evaluating more complete data when it becomes available'. 

Reviewer 2 Report

This manuscript describes the short-term effects of particulate air pollution (PM2.5) on cardiovascular and respiratory diseases in Kosovo, and it draws conclusions by analyzing the electronic database of primary care visits of pediatric clinics, avoiding the interference of numerous factors: old age death, other diseases, etc. Moreover, it also points out some improvements in this research, such as long-term temporal changes and spatial analysis. However, following points need to be clarified in the revised manuscripts before it is considered to be accepted.

1、In order to emphasize the significance of this research, it is also necessary to point out how air pollution causes cardiovascular diseases and respiratory diseases, i.e., why study these two diseases, as well as the harm of diseases. Some representative references are recommended to be cited, such as Adv. Funct. Mater. 2020, 30, 2002434, J Hazard. Mater. 2021, 403, 123910, Environ. Int. 2015, 74, 136-143, J. Med. Toxicol. 2012, 8, 166-175, etc.

2、The introduction is quite lengthy, and the inaccuracy of the database was also mentioned in the conclusions. Therefore, whether to consider deleting this section in the introduction?

3、ICO-10 should be given the full name when it appears for the first time.

4、The coordinates in the figure can not be seen clearly, and the single color can not directly express the results.

5、Please consider whether it is indispensable to explain clearly why examined the effects on respiratory visits on lags 4 through 10.

Author Response

Reviewer 2

This manuscript describes the short-term effects of particulate air pollution (PM2.5) on cardiovascular and respiratory diseases in Kosovo, and it draws conclusions by analyzing the electronic database of primary care visits of pediatric clinics, avoiding the interference of numerous factors: old age death, other diseases, etc. Moreover, it also points out some improvements in this research, such as long-term temporal changes and spatial analysis. However, following points need to be clarified in the revised manuscripts before it is considered to be accepted. We want to express our thanks for the helpful comments.

1、In order to emphasize the significance of this research, it is also necessary to point out how air pollution causes cardiovascular diseases and respiratory diseases, i.e., why study these two diseases, as well as the harm of diseases. Some representative references are recommended to be cited, such as Adv. Funct. Mater. 2020, 30, 2002434, J Hazard. Mater. 2021, 403, 123910, Environ. Int. 2015, 74, 136-143, J. Med. Toxicol. 2012, 8, 166-175, etc. Thank you for the suggestions. We have added several high-level publications summarizing the impact of air pollution on respiratory and cardiovascular health.

2、The introduction is quite lengthy, and the inaccuracy of the database was also mentioned in the conclusions. Therefore, whether to consider deleting this section in the introduction? We moved the limitations to the discussion section.

3、ICO-10 should be given the full name when it appears for the first time. We used the full term when ICD was first mentioned.

4、The coordinates in the figure can not be seen clearly, and the single color can not directly express the results. We increased the letter size for the figures. We do not understand the color issue.

5、Please consider whether it is indispensable to explain clearly why examined the effects on respiratory visits on lags 4 through 10. Well, it was a post-hoc decision. We do find it only fair to acknowledge that. But we also want to point out that these lags are not without biological plausibility.

Round 2

Reviewer 1 Report

Thank you for addressing the questions. 

Author Response

thank you!

Reviewer 2 Report

This manuscript presents the impact of air particle pollution on human cardiovascular and respiratory diseases. The article is clear and complete in structure, and the relevant data is carefully investigated and analyzed. I think this manuscript could be accepted after some revision. The detailed comments are listed following:

1. The data figures in the article lack horizontal and vertical axes.

2. Please check the text carefully. For example, "table" needs to be capitalized.

3. After the experimental results are shown, the importance of developing air filtration protective materials can be properly explained, and the following references are recommended: Adv.Mater. 2020, 32, 200361, J Mater. Chem. A 2020, 8, 18955-18962, ACS Appl. Mater. Interfaces 2019, 11, 2750-2757, etc.

Author Response

I have capitalized the words “Table” and “Figure”, when they referred to a table or figure in the paper.

I do not understand the suggestion regarding the figures. In all figures, the x-axis displays the lags (0-6 or 4-10, as defined on the axis) and the y-axis displays the coefficient per µg/m³ (which is also defined). Which axis is missing? I did draw a bold line at the y-axis though. I hope this satisfies the reviewer.

I do not understand how filtration links to our paper. Does the reviewer suggest outfitting all houses in Kosovo with air filtration? I cannot find the first reference. Article numbers in Adv.Mater. 2020, 32 have one digit more than the 6 in “200361”. And the articles are not consecutively numbered in that journal. Therefore, searching for an article by its number is a very difficult task anyway. I do not have free access to the second paper and therefore can only read the abstract. It is about “filters from biomimetic wet-adhesive nanoarchitectured networks”. This is certainly not my field of expertise! The third paper is about face masks. Yes, highly efficient face masks without much resistance would be beneficial especially when it comes to the prevention of infectious diseases. But with a similar argument a reviewer could suggest citing studies on the effectiveness of vaccination or the benefits of a certain disinfectant.